# An Overview of Different Techniques for Improving the Treatment of Pulmonary Hypertension Secondary in Systemic Sclerosis Patients

**DOI:** 10.3390/diagnostics12030616

**Published:** 2022-03-01

**Authors:** Barbara Ruaro, Francesco Salton, Elisa Baratella, Paola Confalonieri, Pietro Geri, Riccardo Pozzan, Chiara Torregiani, Roberta Bulla, Marco Confalonieri, Marco Matucci-Cerinic, Michael Hughes

**Affiliations:** 1Department of Pulmonology, Cattinara Hospital, University of Trieste, 34149 Trieste, Italy; francesco.salton@gmail.com (F.S.); paola.confalonieri.24@gmail.com (P.C.); pietrogeri@gmail.com (P.G.); riccardo.pozzan@asugi.sanita.fvg.it (R.P.); torricina@gmail.com (C.T.); marco.confalonieri@asugi.sanita.fvg.it (M.C.); 2Department of Radiology, Cattinara Hospital, University of Trieste, 34149 Trieste, Italy; elisa.baratella@gmail.com; 3Department of Life Sciences, University of Trieste, 34149 Trieste, Italy; rbulla@units.it; 4Department of Experimental and Clinical Medicine, Division of Rheumatology, University of Firenze, 50139 Florence, Italy; marco.matuccicerinic@unifi.it; 5Unit of Immunology, Rheumatology, Allergy and Rare Diseases (UnIRAR), IRCCS San Raffaele Hospital, 20132 Milan, Italy; 6Tameside and Glossop Integrated Care NHS Foundation Trust, Ashton-Under-Lyne OL6 9RW, UK; michael.hughes-6@postgrad.manchester.ac.uk; 7Division of Musculoskeletal and Dermatological Sciences, Faculty of Biology, Medicine and Health, The University of Manchester, Salford Royal NHS Foundation Trust, Manchester M1 1AA, UK

**Keywords:** systemic sclerosis right heart catheterization pulmonary arterial hypertension (PAH), interstitial lung disease, high-resolution computed tomography

## Abstract

In systemic sclerosis (SSc) mortality is mainly linked to lung involvement which is characterized by interstitial lung disease (ILD) and pulmonary hypertension (PH). In SSc, PH may be due to different etiologies, including ILD, chronic thromboembolic disease, pulmonary veno-occlusive disease, and pulmonary arterial hypertension (PAH). The main tool to screen PAH is transthoracic echocardiography (TTE), which has a sensitivity of 90%, even if definitive diagnosis should be confirmed by right heart catheterization (RHC). The radiological evaluation (i.e., HRTC) plays an important role in defining the possible causes and in monitoring the evolution of lung damage. For PAH, identifying individuals who have borderline elevation of pulmonary arterial pressure needs to be appropriately managed and followed. In the past few years, the strategy for the management of PAH has significantly evolved and new trials are underway to test other therapies. This review provides an overview of the tools to evaluate PAH in SSc patients and on treatment options for these patients.

## 1. Introduction

Systemic sclerosis (SSc) is a connective tissue disease characterized by immune and endothelial dysfunction, inflammation and fibrosis [1,2,3,4,5,6,7,8]. The main cause of death in SSc patients is the effect that collagen overproduction has on the internal organs, such as the pulmonary system [9,10,11,12]. Lung involvement in SSc patients includes fibrosis or pulmonary hypertension (PH) [1,13,14,15,16,17,18,19,20]. In 2018 it was proposed to lower the cut-off for the diagnosis of PH to a mean pulmonary artery pressure (mPAP > 20 mmHg) [12,13,14,21,22,23,24,25,26,27]. PH can be classified as pre- and post-capillary, and more precisely pre-capillary PH is characterized by a pulmonary artery wedge pressure (PAWP) ≤ 15 mmHg and a pulmonary vascular resistance (PVR) ≥ 3 WU, while post-capillary PH is defined by a PAWP > 15 mmHg with normal PVR [12,13,14]. The increase in afterload due to a chronic elevation of pressure in the pulmonary circulation can progressively affect the right ventricle (RV), leading to RV dysfunction [10,11,12,13,14,15,16]. Five categories have been proposed, on the basis of clinical characteristics and pathophysiological mechanisms: Group 1 (PAH); Group 2 (PH due to left heart disease); Group 3 (PH due to lung disease or hypoxia); Group 4 (chronic thromboembolic pulmonary hypertension (CTEPH)); and Group 5 (PH due to unclear or multi-factorial mechanisms). Patients in Group 1, Group 3 and Group 4 present a pre-capillary form of PH, while those in Group 2 may present both an isolated post-capillary PH or a combined PH with a pre-capillary component, given by an elevated diastolic pressure gradient or an increase in pulmonary vascular resistance [10,11,12,13,14,15,16]. In SSc, PH can be correlated by: vasculopathy of the small pulmonary arteries (Group 1 PAH); interstitial lung disease (Group 3); and left ventricular systolic or diastolic dysfunction (Group 2) [10,11,12,13,14,15,16]. In SSc, pulmonary hypertension phenotyping should be achieved to provide a correct treatment regimen [10,11,12,13,14,15,16,17,18,19,20]. It is likely that PAH (classified by the WHO as group I) is present in a consistent number of SSc patients, but its mixed forms (either WHO group III with interstitial lung disease) also exist, while some patients may also have post-capillary PH (WHO group II) [10,11,12,13,14,15,16,21,22,23]. Pulmonary hypertension associated with ILD generally derives from an involvement of more than >20% of the lung parenchyma [10,11,12,13,14,15,16,21,22,23]. Some patients may also develop Pulmonary Veno-Occlusive Disease (WHO group 1) or Chronic Thrombo-embolic PH (WHO group 4) (Table 1) [10,11,12,13,14,15,16,21,22,23]. However, PH frequency varies across different reports depending upon the duration of follow-up [10,11,12,13,14,15,16,23,24,25,26,27,28,29,30,31,32,33,34].

The SSc limited cutaneous subset has been associated with PAH [10,11,12,13,14,15,16,23,24,25,26,27,28,29,30,31,32,33,34], and develops in approximately 10% of individuals with SSc [14,21,22,23]. These patients are more likely to have severe intrinsic right ventricular dysfunction, high BNP (B-type natriuretic factor), low DLCO and poor survival [10,11,12,13,14,15,16,23,24,25,26,27,28,29,30,31,32,33,34]. Lungs from patients with SSc-PAH present a characteristic vascular pathology where perivascular lymphocytes and intimal fibrosis are the hallmarks [10,11,12,13,14,15,16,23,24,25,26,27,28,29,30,31,32,33,34,35]. The complex clinical presentation and the distinct vascular pathology of SSc-PAH suggest a unique biology among group I PH conditions [10,11,12,13,14,15,16,23,24,25,26,27,28,29,30,31,32,33,34]. There have been increasing efforts to understand the relationship between autoimmunity, inflammation and the evolution of PAH [10,11,12,13,14,15,16,23,24,25,26,27,28,29,30,31,32,33,34]. The role of autoimmunity has also suggested the use of immunosuppressors as potential therapeutic candidates after standard-of-care vasoactive and vasodilating therapy [16,34]. 

This review provides an overview on the screening and diagnostic tools and on the treatment options for PAH in SSc patients.

## 2. Screening Procedures in SSc

Recommendations and systematic reviews have proposed different methods of screening and follow-up of PAH. In particular, transthoracic echocardiography, natriuretic peptide NT-proBNP, and spirometry are suggested for screening/follow up while right heart catheterization is reserved for diagnosis (Table 2).

### 2.1. Transthoracic Echocardiography

Transthoracic echocardiography (TTE) is the gold standard for the screening of PAH in SSc patients. In ESC/ERS Guidelines and in literature, a TTE is recommended every year in SSc patients, with or without symptoms [13,14]. The usual parameters used to determine the probability of developing PH are: the tricuspid valve regurgitation velocity (TRV), the dilation of the right section of the heart, the presence of pericardial effusion and the dilation of the inferior vena cava. The 2D-speckle tracking echocardiography estimates the strain on the right atrium (RA) and the right ventricle (RV) of the heart [13,14]. Recently a study demonstrated that a deformity of systolic longitudinal peak of 14.48% at the apical segment of the lateral wall of RV has a 100% specificity for PAH development in SSc [13,14]. However, echocardiography has a low predictive value (e.g., the ultra-sound method is operator-dependent).

### 2.2. Right Heart Catheterization

Right heart catheterization (RHC) is the best technique for PAH diagnosis [10,11,12,13,14,15]. The RHC provides useful information on the degree of hemodynamic damage, determines response to treatments and establishes prognosis of PAH [19,33,34,35,36,37,38,39,40,41,42,43]. The RCH evaluation is recommended when patients have an intermediate or high risk of developing PH, based on ETT evaluation, defined as a TRV peak >2.8 ms^−1^, or a TRV <2.8 ms^−1^ (or not measurable) with other variables suggestive of PH [38]. 

The assessment of pulmonary arterial wedge pressure could be over- or under-estimated (Figure 1). In fact, there are no internationally accepted clinical guidelines presenting the best practice for performing RHC. However, it has also been proposed that the standardization of RHC should be completed, to optimize the use of this technique in routine clinical practice [10,11,12,13,14,15].

### 2.3. Spirometry

In SSc-PAH patients, there is a frequent reduction in the diffusion of lung carbon monoxide (DLCO) with normal forced vital capacity values (FVC) [26,27]. The reduction of DLCO during the follow-up of SSc patients could be considered as a sign of possible PAH presence [26,27]. A DLCO value < 50% has a high specificity and positive predictive value, but a normal DLCO value does not exclude the presence of PAH [26,27]. A more accurate assessment can be obtained by evaluating the FVC%/DLCO% ratio, as suggested in the DETECT algorithm [23,26,27].

### 2.4. Radiological Tools

Radiological evaluation plays an important role in defining the possible causes and in monitoring the evolution of the disease.

#### 2.4.1. Chest X-ray

The chest X-ray can show classical radiological findings of PH only in the late stage of the disease. Typical radiological findings are represented by central pulmonary arterial dilation, increase of the diameter of the right interlobar artery (>16 mm) and reduction of the retrosternal space related to right ventricular enlargement [36]. Moreover, chest X-ray can be useful in underlying pulmonary parenchymal disease.

A CT scan is more accurate than chest X-ray in recognizing the radiological features of pulmonary hypertension and a systematic approach is recommended in evaluating CT scans, to promptly recognize parenchymal, vascular and cardiac signs of pulmonary hypertension [37].

#### 2.4.2. High-Resolution Computed Tomography (HRTC)

*Parenchymal signs.* There are different forms of pulmonary involvement in SSc. HRCT plays a central role in the diagnosis of ILD (interstitial lung disease), in recognizing the pattern of lung involvement and in providing information regarding the extension of the interstitial involvement [3] (Figure 2).

It is possible to distinguish two different phenotypes of patients with SSc based on the association between ILD and pulmonary hypertension: the first one occurs when PH is associated with ILD, the second one when PH is not related with the extension of the ILD [38]. The association of ILD and PH increase the mortality risk to five-fold in SSc-PAH [39].

Vascular signs. The pulmonary vascular involvement in SSc can cause PAH. Typical radiological findings are represented by a main pulmonary artery with a diameter equal of greater than 29 mm measured in a scanning plane of its bifurcation, this sign has a positive predictive value of 97%, sensitivity of 87% and specificity of 89% for the presence of PH [37,40,41]. 

Meanwhile a diameter of the main pulmonary artery that is greater than that of the ascending aorta (PA/AA > 1) is another vascular sign of pulmonary hypertension, with a appositive predictive value of 96% and a specificity of 92% [42]. 

#### 2.4.3. Classification in Different WHO Group of Pulmonary Hypertension

HRCT and CT angiography are particularly useful to subdivide patients into different groups, according to underlying causes of pulmonary hypertension (HP). 

Interstitial lung disease (WHO Group III of pulmonary hypertension). ILD is present in up to 90% of patients with SSc [43]. The most common ILD in the SSc patient is the NSIP pattern, characterized by a different degree of inflammation and fibrosis (cellular and fibrotic NSIP). The cellular NSIP pattern is characterized by the presence of ground glass opacities with a bilateral and symmetrical distribution; when fibrotic changes occur (fibrotic NSIP pattern), irregular septal thickening, bronchiectasis and bronchiolectasis can be recognized on HRCT as superimposed features on a background of pre-existing ground glass opacities [43]. The usual interstitial pneumonia (UIP) pattern is less common in SSc and this pattern is characterized by the presence of irregular septal thickening in the peripheral and subpleural regions of the lungs, honeycombing, traction bronchiectasis and pulmonary volume loss. This pattern has a worst prognosis and a faster progression [3] (Figure 3).

PH can be detected in patients with ILD, in particular in those patients with an extension of fibrotic changes greater than 20% of the lung volume [3], and this is an indicator of morbidity and mortality [3].

Combined pulmonary fibrosis and emphysema syndrome (WHO Group III of pulmonary hypertension). This is a relatively new entity characterized by the presence of emphysema, both paraseptal and centrilobular, in upper lobes in association with lung fibrosis in lung bases. It occurs most often in smokers. This syndrome is associated with a higher risk of developing pulmonary hypertension and is described in patients with connective tissue diseases, including SSc patients [44]. Moreover, it can be identified in SSc patients with no history of smoking exposure [45].

Pulmonary veno-occlusive disease (WHO Group I of pulmonary hypertension). This rare condition is characterized by a diffuse obstruction of the small pulmonary veins. Typical HRCT findings are represented by the presence of centrilobular ground glass opacities, septal thickening, and lymph nodes enlargement. POVD-like aspects are common in patients with SSc-PH [46].

SSc patients with pulmonary hypertension and radiological signs of POVD have a worst prognosis compared with those patients without them therefore it is important to distinguish this phenotype not only at the baseline CT examination but also during follow-up [47]. 

Chronic thromboembolic pulmonary hypertension (WHO Group IV of pulmonary hypertension). SSc patients are at higher risk of developing chronic thromboembolic disease, and this form of PH that can be surgically treated. Typical vascular signs of CTEPH can be detected on CT angiography, including vascular stenosis, retraction with total or subtotal obstruction, recanalization or residual bands within the vascular lumen. Dilatation of systemic supply is another feature, which is the abnormal enlargement of bronchial, intercostal, phrenic and internal mammary arteries [48].

Cardiac features of PH are represented by a right ventricle hypertrophy, defined as a wall thickening of more than 4 mm or leftward bowing of the interventricular septum, and right ventricular dilatation, defined as a right ventricle to left ventricle diameter ratio of more than 1:1 at the midventricular level on axial image. [49].

Moreover, linear bands, due to previous pulmonary infarction, mosaic pattern of perfusion, bronchial wall thickening and cylindrical airway dilatation (segmental and subsegmental bronchi) are typical parenchymal features of chronic pulmonary hypertension detectable on CT scan [37,50].

#### 2.4.4. Cardiac Magnetic Resonance Imaging

Cardiac involvement is very common in SSc patients. The fibrotic myocardiopathy can cause a dysfunction of the left heart and then an increase of the risk of arrhythmic and ischemic complications, associated with the development of post-capillary pulmonary hypertension (group 2 of the diagnostic classification) [10,11,12,13,51]. The myocardial fibrosis is observed in 50–80% of post-mortem cases [10,11,12,13,52].

The structural alterations are better visualized on cardiac magnetic resonance imaging (MRI) [51]. MRI can also demonstrate pericardial effusion [52]. Several studies have demonstrated that cardiac MR is a very sensitive procedure for the evaluation of myocardial fibrosis in SSc patients and can help to distinguish group 1 (PAH) and group 2 pulmonary hypertension [10,11,12,13]. At the time of writing, the role of cardiac MRI in the evaluation of PAH is not well established. 

### 2.5. Biomarkers

The natriuretic peptide (NT-proBNP) is the most frequent used biomarker in SSc, but it is not specific for right ventricular dysfunctions [19,33,34,35,36,37,38,39,40,41,42,43]. This biomarker may be useful in combination with other screening techniques and in screening algorithms [19,33,34,35,36,37,38,39,40,41,42,43]. The increase of serum levels of uric acid has been included in the DETECT algorithm [19,33,34,35,36,37,38,39,40,41,42,43]. Further studies are still needed to demonstrate the utility of blood markers in PAH diagnosis. 

## 3. Screening Algorithms

The use of algorithms within SSc-PAH screening is associated with a better outcome. Here, we describe three (Table 2) of the most widely used algorithms for PAH screening in SSc [37,53,54,55,56,57,58,59,60].

ESC/ERS Guidelines recommend early echocardiographic screening in asymptomatic SSc patients, followed by the measurement of biomarkers and DLCO [14].

DETECT algorithm is based on two steps: first, the evaluation of a group of functional and blood parameters related to high risk; in case of positivity with a high-risk score, the patient undergoes the second step which evaluates echocardiography parameters. If the second step score is still elevated, the patient will then receive a RHC [23].

ASIG is based on two steps, evaluating DLCO, FVC/DLCO and NT-proBNP values. If either or both parameters (step A) are present patients must then undergo echocardiography associated with other confirmatory tests like HRTC and 6 MWT (step B). If there is no parenchyma lung involvement, a RHC should be performed on the patient [19,37,53,54,55,56,57,58,59].

### Impact of Screening for SSc-PAH

There are encouraging data to support the fact that survival from SSc-PAH has significantly improved over time, including through the detection of milder disease. Furthermore, new and combination therapies (discussed later) have dramatically changed the treatment paradigm of SSc-PAH. In the study by Humbert et al., which included two cohorts of SSc-PAH from the same management era, patients enrolled in a systematic PAH evaluation program had less advanced pulmonary vascular disease compared to those who were symptomatic and diagnosed by SSc-PAH by RHC [61]. Furthermore, survival rates were considerably higher in patients who were identified by the PAH detection evaluation at 1-year (100% vs. 75%), 3-years (81% vs. 31%), 5-years (73% vs. 25%) and 8-years (64% vs. 17%) [61]. Similarly, in a recent meta-analysis of randomized, controlled trials (n = 11) and observational registries (n = 19), the survival rate of CTD-PAH patients treated after (compared to before) 2010 was higher at 3 years (73% vs. 65%) [62]. The authors also highlighted that, although survival in CTD-PAH has improved over the last 10 years, the risk of death was still higher than in those with PAH overall [62]. However, in a recent meta-analysis study examining trends in vascular disease in SSc over time, there was a significant improvement in PAH 1-year mortality, but not in PAH 3-year or 5-year mortality [63]. The authors speculated that it will likely require more time to assemble enough treated patients to determine improvement in outcomes. Furthermore, changes in treatment (e.g., upfront combination therapy) are relatively recent and therefore could not translate into the results of the meta-analyses. The authors also observed that there was trend for improvement in three surrogate measures (mPAP, PVR and RAP) of SSc-PAH severity, which supports that regular screening for PAH is an effective strategy to identify and likely treat patients with milder disease [63]. 

## 4. Treatment of SSc-PAH

### 4.1. Overview of SSc-PAH Treatment

Patients should be managed by expert specialist PAH centers. Many valuable authoritative reviews have been written on the management of PAH and guidelines developed by the ESC/ERS for the diagnosis and treatment of PAH [14,64]. Giordano et al. [65], review the relevance and impact of the 2015 ESC/ERS guidelines to patients with SSc-PAH. In this review we will focus on the major drug classes, including combination therapy, relevant to SSc-vasculopathy in SSc. Of interest, a unified vascular endophenotype has been proposed in SSc where vascular-acting therapies could be deployed with a potential disease-modifying effect before the onset of irreversible tissue fibrosis and organ dysfunction [66,67,68].

### 4.2. Principes of the Treatment Strategy of PAH

The ESC/ERS guidelines divide the treatment strategy of PAH into three steps [14]. First is general measures and supportive therapies (oxygen, anticoagulation, diuretics and digoxin) [14]. Patients should be referred to expert specialist centers and undergo acute vasoreactivity testing for an indication of the need for treatment with calcium channel blockers. The second step consists of calcium channel blockers in vasoreactive patients and PAH-approved therapies in non-vasoreactive patients, either as mono- or combination-therapy according to the prognostic risk [14]. The third step is dictated by the response to the initial treatment strategy. If there is an inadequate response, then combination therapy or lung transplantation should be considered [14]. 

### 4.3. General Measures

Patient education is essential and should be delivered by a dedicated multi-disciplinary team. Patients require high quality and appropriate education throughout the course of their disease [69]. General measures include (but are not limited to) supervised exercise training and rehabilitation, avoiding pregnancy, immunizations (influenza and pneumococcal) and psychosocial support [14]. In elective surgery, an epidural is preferred to general anesthesia whenever possible [14].

### 4.4. Drugs Therapies for SSc-PAH

The target mechanisms, drug classes and examples of the major therapies used for the treatment of SSc-PAH are presented in Table 3.

#### 4.4.1. Nitric Oxide Pathway

Nitric oxide is an endogenous vasodilator and also has anti-proliferative effects. Inhibition of the cyclic guanosine monophosphate (cGMP) degrading enzyme phosphodiesterase type-5 (PDE5) results in vasodilation through increasing the local bioavailability of nitric oxide. PDE5 inhibitors approved for the treatment of SSc-PAH are sildenafil (SUPER-1 and SUPER-2 trials) [70,71] and tadalafil (PHIRST-1 and PHIRST-2 trials) [72,73]. Riociguat is an approved guanylate cyclase stimulator which augments the NO-cGMP pathway [74]. The combination of PDE5 inhibitors and riociguat is strictly contraindicated due to the potential for significant hypotension.

#### 4.4.2. Prostacyclin Pathway Agonists

Prostacyclin is a powerful vasodilator and also inhibits platelet aggregation and cell proliferation. Prostacyclin analogues can be delivered by different routes of administration: intravenously, subcutaneous, orally and via inhalation. The drugs include Iloprost, epoprostenol, treprostinil which are all approved for the treatment of SSc-PAH. Currently, intravenous iloprost is only available in Europe (and not the United States). Selexipag is an oral selective IP prostacyclin receptor-agonist. In a phase 3, event-driven randomized, double-blind, placebo-controlled (‘GRIPHON’) trial, which included patients with SSc, the risk of death or complications related to PAH was significantly lower with selexipag [75].

#### 4.4.3. Endothelin Receptor Antagonists

Endothelin-1 is a potent vasoconstrictor, including within the pulmonary vasculature and smooth muscle mitogen. Bosentan (BREATH-1 trial), macitentan (SERAPHIN trial) and ambrisentan (ARIES 1 and 2 trials) are endothelin receptor-antagonists which are approved for the treatment of SSc-PAH [76,77,78,79,80].

#### 4.4.4. Investigational Therapies

Considering the poor prognosis and complex pathobiology of SSc-PAH, other novel therapeutic targets are being actively investigated including immunosuppressive/modulatory therapies (e.g., mycophenolate mofetil) [81]. For example, in a multicenter, double-blind, randomized, placebo-controlled, proof-of-concept trial (n = 57), B-cell depletion with rituximab (1000 mg administered 2 weeks apart) was found to be safe and effective, and, although the primary end-point (mean of change in 6 MWD) favored treatment, this did not reach statistical significance [82].

## 5. Conclusions

In conclusion, many conditions can cause pulmonary hypertension in patients with SSc, and sometimes different causes can co-exist in the same patient, therefore all the tools to evaluate the damage play an important role in defining the possible causes and in monitoring the evolution of the disease. The correct and early elevation of pulmonary arterial pressure is essential to decide the appropriate management and follow-up. In fact, recently, there was an increase in strategies for the management of PAH and other potential therapies. Furthermore, new drugs are being studied in several trials to provide robust evidence of their role in the improvement of long-term SSc-PH outcomes. 

## Figures and Tables

**Figure 1 diagnostics-12-00616-f001:**
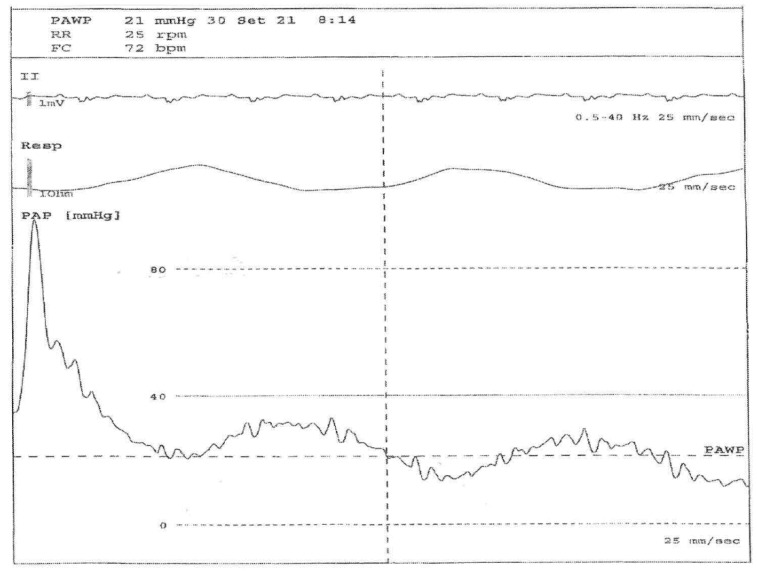
Right heart catheterization (RHC) evaluation in a 56-year-old systemic sclerosis patient with pulmonary arterial hypertension (PAH). Legend: From below: pulmonary arterial pressure, respiratory and EKG waveforms during arterial catheterization. The first part of the pressure trace reflects the pressure in a pulmonary artery (large swings, dicrotic notch), then the balloon is inflated and the tip of the Swan–Ganz catheter floats until it wedges in a small artery (small swings synchronous with respiratory rate), allowing a pulmonary arterial wedge pressure (PAWP) to be obtained, which is an indirect measure of left ventricle pressure.

**Figure 2 diagnostics-12-00616-f002:**
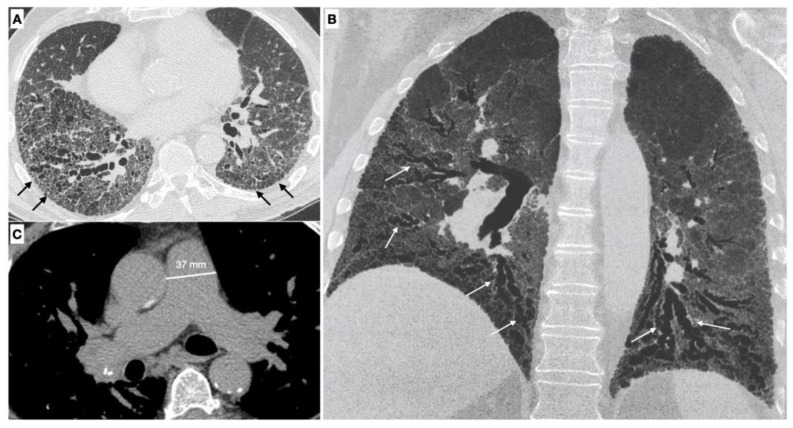
(**A**,**B**) Parenchymal signs: axial high resolution CT scan shows a fibrotic NSIP pattern due to the presence of diffuse irregular septal thickening (black arrows), bronchiectasis and bronchiolectasis on a background of diffuse ground glass opacities. Both lower lobes are symmetrically involved by these alterations. Bronchiectasis and bronchiolectasis (white arrows) are better visualized on the MinIP coronal plane reconstruction; (**C**) Vascular signs: a main pulmonary artery with a diameter greater than 36 mm measured in a scanning plane of its bifurcation is a sign of pulmonary hypertension.

**Figure 3 diagnostics-12-00616-f003:**
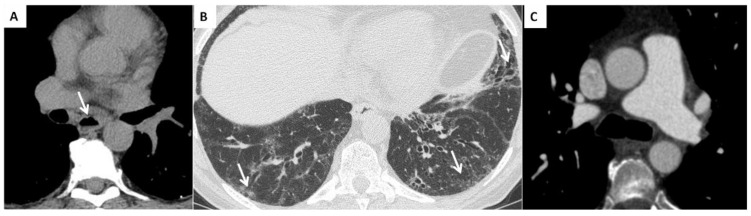
(**A**) Esophageal involvement in a 56-year-old male with a diagnosis of SSc. Axial image with a mediastinal window setting shows a dilatation of the esophagus (diameter >1.2 cm) with an air-fluid level (white arrows); (**B**) Parenchymal signs: axial high resolution CT scan shows a fibrotic NSIP pattern due to the presence of diffuse irregular septal thickening, bronchiectasis and bronchiolectasis on a background of diffuse ground glass opacities (white arrows); (**C**) Vascular signs: a main pulmonary artery with a diameter greater than 36 mm measured in a scanning plane of its bifurcation is a sign of pulmonary hypertension.

**Table 1 diagnostics-12-00616-t001:** Different phenotypes of PH in SSc patients.

Differences in PH in SSc Patients	Epidemiology and Features	Risk Factors	Diagnosis	Therapy	Outcome
PAH group 1 (pulmonary arterial hypertension)	The overall PAH prevalence found was 6.4% (95%CI 5–8.3%) [15]	Increased age	Intrinsic RV contractile function and reduced RV contractile reserve	- PAH-specific medication often in combination - PDE5i are first-line therapy	- Worse prognosis than IPAH - Patients with IPAH die for PH-related complications
PAH group 1′ (pulmonary veno-occlusive disease, PVOD)	Rare form up to 15% of patients with SSc-associated PH might have elements of PVOD Rarely observed as pure cause of PH		- Similar hemodynamics to PAH - CT findings of septal lines, nodules and lymphadenopathy	- Reduced response to supportive measures and pulmonary vasodilators - Immunosuppressive therapy might be used	Poor prognosis
PAH group 2 (correlated with left heart diseases)	Left ventricular dysfunction due to CAD, arrhythmias, HFpEF and HFrEF	- Arterial hypertension, obesity, diabetes and atrial fibrillation, typically found in patients with HFpEF - Left heart disease can be related to older age, primary myocardial fibrosis, smoking or renal disease and hypertension	Postcapillary PH at RHC (PAWP > 15 mmHg) in PH with left heart disease	- Vasodilator with PAH-specific therapies are useful - PDE5i were selected as first-line therapy	- Worse prognosis in case of HFpEF - Elevated PAP and RV dysfunction - are predictors of death in patients with HFpEF
PAH group 3 (associated to lung diseases)	From 15 to 50% SSc patients present with ILD	- Diffuse cutaneous systemic sclerosis - Severe gastro-esophageal reflux - Antibodies (anti-DNA topoisomerase 1, anti-Th/To ribonucleoprotein (RNP), - anti-U11/U12 RNP) - Epithelial damage (elevated mucin 1 levels or fast diethylenetriamine pentaacetate clearance)	- Difficult classification - HRCT, PFT and DLCO data are useful to identify and in staging ILD as a cause of Group 3 PH - >20% fibrotic lung volume - involvement (by HRCT) and/or FVC < 70% at PFT	- >90% of patients with PH-HFpEF receive PDE5i - ERAs are rarely used - These therapies might aggravate hypoxia and V/Q ratio - Immunosuppressive therapy and/or nintedanib are options	Worst prognosis in ILD-PH
PAH group 4 (chronic thromboembolic PH)	Intravascular thrombosis sometimes is present in SSc-PAH	Anti-phospholipid antibodies increase risk of PTD	- CTEPH should be investigated because can treated surgically - V/Q scan is diagnostic	- Surgical treatment is recommended with medical - therapy and balloon pulmonary angioplasty - Rociguat can be used in inoperable CTEPH	
PAH group 5 (unclear and/or multifactorial mechanism)	Rare causes of PH coexisting with SSc Frequency depends on involved mechanisms	Multiple mechanisms might be considered as part of this group	Routine investigation and extensive additional tests to confirm multifactorial causes		

Legend: SSc, systemic sclerosis; IPAH: idiopathic pulmonary arterial hypertension; PH, pulmonary hypertension; PAH: pulmonary arterial hypertension; PTD: pulmonary thromboembolic disease; CTEPH, chronic thromboembolic pulmonary hypertension; ERAs, endothelin receptor antagonists; RV, right ventricle; PVOD, pulmonary veno-occlusive disease; CAD, coronary artery disease; HFpEF, Heart failure with preserved ejection fraction; HFrEF, heart failure with reduced ejection fraction; ILD, interstitial lung disease; HRCT, high resolution computed tomography; FVC, forced vital capacity; PFT: pulmonary function tests; DLCO: diffusion lung CO.

**Table 2 diagnostics-12-00616-t002:** Investigation tools to properly categorize different types of systemic sclerosis–pulmonary hypertension (SSc–PH).

Procedures	Skills	Drawbacks
Transthoracic echocardiography (TEE)	- Most frequently used test for screening of SSc-PAH - Recommended in all SSc patients - Monitoring TRV and indirect signs of PH - RHC is recommended in case of intermediate-high risk of PH	- Low predicted value
Tests of lung function	- Reduction of DLCO (<50%) with normal FVC has high specificity (90%) and high positive predictive value for SSc-PAH - Predicted DLCO > 60% excludes PAH	- DLCO reduction is associated also with pulmonary fibrosis and obstructive syndrome - Conflicting results between studies evaluating DLCO components
Cardiopulmonary exercise testing	- A reduction in peak oxygen consumption and an increase in the ratio of ventilation to CO2 production are frequently observed	- Poor precision of PAPs measurement and cardiac output during exercise does not recommend the test for screening
Cardiac magnetic resonance imaging	- Essential procedure in evaluation of the heart in SSc patients (cardiac involvement and myocardial fibrosis) - Sensitive procedure for identifying myocardial fibrosis in patients with SSc and able to distinguish group 1 and group 2 pulmonary hypertension	
Biomarkers	- NT-proBNP most used biomarker in SSc - Useful biomarker in combination with other screening tests such as echocardiography and PFT - Red blood cell distribution width (RDW) and high serum levels of uric acid are associated with a good prediction and increased risk of SSc-PAH - Anti-centromere antibodies (ACA) are associated with risk of developing PAH - Anti-SCL70 antibodies seem to be protective	- NT-proBPN inadequate for screening for low sensitivity, low negative predictive value, and possible impairment in patients with left heart failure
Screening algorithms	- ESC/ERS Guidelines, DETECT algorithm and ASIG algorithm are the most widely used - ESC/ERS Guidelines recommend early echocardiographic screening in asymptomatic patients with SSc, followed by the evaluation of biomarkers and DLCO - DETECT algorithm is based on 2 steps: 1^ evaluation of FVC/DLCO, presence of telangiectasias, ACA positivity, values of NT-proBNP and uric acid, right axial deviation in ECG. If the total risk score of the first step is >300 points, then 2^ step which evaluates echocardiography variables. If the 2^ step score is >35 the patient will perform RHC. - ASIG is based on two steps: 1^, with DLCO < 70% and FVC/DLCO value > 1.8, and 2^, with NT-proBNP values > 210 pg/mL. The screening is positive if either or both are present. In this last case patients must undergo echocardiography associated with other confirmatory tests like HRTC and 6 MWT. If there is no parenchyma lung involvement, RHC should be performed.	

Legend: PAH: pulmonary arterial hypertension; PH: pulmonary hypertension; mPAP: mean pulmonary artery pressure; HFpEF, Heart failure with preserved ejection fraction; HFrEF, heart failure with reduced ejection fraction; RHC: right heart catheterization; PAWC: pulmonary artery wedge pressure; PVR: pulmonary vascular resistance; WU: wood units; RV: right ventricle; STE: speckle-tracking echocardiography; HRCT: high-resolution computed tomography; PVOD: pulmonary veno-occlusive disease; PFT: pulmonary function tests; DLCO: diffusion lung CO; V/Q: ventilation/perfusion; EKG: electrocardiogram; CMR: cardiac magnetic resonance; ILD; interstitial lung disease; FVC: forced vital capacity; CTEPH: chronic thromboembolic pulmonary hypertension; 6 MWT: 6 min walking test.

**Table 3 diagnostics-12-00616-t003:** Target mechanisms, drug classes and examples of therapies used for the treatment of SSc-PAH.

Target Mechanism	Drug Classes Including Examples of Therapies	
Nitric oxide pathway	PDE type-5 inhibitors	Sildenafil
		Tadalafil
	Guanylate cyclase stimulator	Riociguat
Prostacyclin pathway agonists	Prostacyclin analogues	Iloprost
		Epoprostenol
		Treprostinil
	Selective IP prostacyclin-receptor agonist	Selexipag
Endothelin-1	Endothelin receptor antagonists	Bosentan
		Macitentan
		Ambrisentan

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
