# Peer review of "An Overview of Different Techniques for Improving the Treatment of Pulmonary Hypertension Secondary in Systemic Sclerosis Patients"

_diagnostics, 2022, doi:10.3390/diagnostics12030616_

Round 1

Reviewer 1 Report

the paper deals  a quick and wide overview of the diagnosis of pulmonary arterial hypertension
The paper illustrates the reader to a wide and well organised overview about the diagnosis and monitoring of pulmonary arterial hypertension

Author Response

I would like to thank the reviewer for all the comments

Reviewer 2 Report

The report of Ruaro B. et al. is well written and explores deeply the  therapies in patients affected by systemic sclerosis relative to pulmonary hypertension.

Major points to address:

  • Table 1. Even the table is detailed, it would be more comprehensive with more coincise and schematic description (for example first line Therapy: PDE5i are the first line......should be PDE5i in : -77% typical IPAH p.; 81% atypical...).. Outcome: Die for PH-related complication .. and so on...
  • Table 2 : as table 1, table 2 should be more schematic
  • Paraghaph 2 Screening procedure: line 2 WHILE should move after in particular.....after catheterization is RESERVED (USED, UTILIZED)  FOR DIAGNOSIS

Author Response

I would like to thank the reviewer for all comments 

I revised the table and I corrected the error in paragraph 2

Reviewer 3 Report

This is an elegantly written MS. No suggestions!

Author Response

I would like to thank the reviewer for the comment!